# Superior Radiation Resistance of ZrO_2_-Modified W Composites

**DOI:** 10.3390/ma15061985

**Published:** 2022-03-08

**Authors:** Bo Cui, Chunyang Luo, Xiaoxi Chen, Chengqin Zou, Muhong Li, Liujie Xu, Jijun Yang, Xianfu Meng, Haibin Zhang, Xiaosong Zhou, Shuming Peng, Huahai Shen

**Affiliations:** 1Institute of Nuclear Physics and Chemistry, China Academy of Engineering Physics, Mianyang 621900, China; cuibo13@163.com (B.C.); luochunyang@163.com (C.L.); xiaoxichen@126.com (X.C.); chenqinzou1990@163.com (C.Z.); muhongli@caep.com (M.L.); mengxf777@163.com (X.M.); hbzhang@caep.cn (H.Z.); 2Henan Engineering Research Center for Wear of Materials, Henan University of Science and Technology, Luoyang 471003, China; wmxlj@126.com; 3Key Laboratory of Radiation Physics and Technology of Ministry of Education, Institute of Nuclear Science and Technology, Sichuan University, Chengdu 610064, China; jjyang@scu.edu.cn

**Keywords:** W alloys, irradiation effect, microstructure evolution, mechanical properties

## Abstract

The microstructure and mechanical properties of pure W, sintered and swaged W-1.5ZrO_2_ composites after 1.5 × 10^15^ Au^+^/cm^2^ radiation at room temperature were characterized to investigate the impact of the ZrO_2_ phase on the irradiation resistance mechanism of tungsten materials. It can be concluded that the ZrO_2_ phase near the surface consists of two irradiation damage layers, including an amorphous layer and polycrystallization regions after radiation. With the addition of the ZrO_2_ phase, the total density and average size of dislocation loops, obviously, decrease, attributed to the reason that many more glissile 1/2<111> loops migrate to annihilate preferentially at precipitate interfaces with a higher sink strength of 7.8 × 10^14^ m^−^^2^_._ The swaged W-1.5ZrO_2_ alloys have a high enough density of precipitate interfaces and grain boundaries to absorb large numbers of irradiated dislocations. This leads to the smallest irradiation hardening change in hardness of 4.52 Gpa, which is far superior to pure W materials. This work has a collection of experiments and conclusions that are of crucial importance to the materials and nuclear communities.

## 1. Introduction

Exploring nuclear fusion energy has important strategic and economic significance for sustainable development. The development of fusion reactor materials is one of the key challenges for the commercialization of fusion energy [1,2,3]. Particularly, plasma facing materials (PFMs), used as a material that directly contacts the fusion plasma D/T/He, need higher requirements for overall performance [4]. During the steady-state operation of the plasma, the energy acting on the first wall materials is about 100–500 eV, the beam density is about 10^19^~10^20^ m^−2^ s^−1^, the neutron flux is about 10^17^ m^−2^ s^−1^ and the heat load is about 0.5 MWm^−2^ [5]. Tungsten (W) is considered to be the most promising PFMs because of its high melting point, high thermal conductivity, high strength, low physical sputtering rate and low hydrogen retention [4,6,7,8]. The International Thermonuclear Fusion Experimental Reactor (ITER), Fusion Demonstration Power Station (Demo) and China Fusion Engineering Fusion Reactor (CFEFR) all list tungsten as one of the first wall materials candidate. Therefore, the studies on irradiation damage of tungsten materials are important issues for fusion reactor service performance.

Over the past decades, extensive microstructural characterizations of W alloys have already been carried out to understand the microstructural response to neutron irradiation [7,9,10,11,12]. It can be concluded that neutron irradiation can cause the formation of lattice defects such as dislocation loops, Frenkel pairs and vacancies in the W alloys [13,14], and can also lead to the formation of rhenium-(Re)- or osmium-(Os)-rich precipitates due to the trigger transmutation process [15,16]. However, owing to low irradiation intensity, long experimental periods, radioactive problems and resource shortages, researchers mostly used different kinds of heavy ion irradiation to screen tungsten materials because of the more obvious damage effect as well as the flexible control for experiment parameters [17,18]. Yi et al. [19] studied that the effects of heavy ion fluence and temperature on dislocation loops size and density and then thoroughly investigated the configurations, geometries and distributions on dislocation loops with different Burgers vector types. For instance, in all cases the damage comprised dislocation loops, mostly of the interstitial type, with Burgers vectors b = 1/2<111> (>60%) and b = <100> by 2 MeV W^+^ radiation to doses 3.3 × 10^17^–2.5 × 10^19^ W^+^/m^2^ at temperatures ranging from 300 to 750 °C. With increasing irradiation temperature, the loop size distributions shifted towards larger sizes, and there was a substantial decrease in loop number densities. The damage microstructure was less sensitive to dose than to temperature [20]. El-Atwani [21] also concluded damage stages as function of irradiation temperature during heavy ion irradiation and demonstrated the influence of the grain size and irradiation dpa rate on dislocation distribution at different irradiation temperatures. In addition, in situ TEM was used to directly observe dynamic response for defect densities dependent on irradiation dose and fluence in the ultrafine and nanocrystalline W alloys. It can be confirmed that at all irradiation temperatures, the evolution of damage microstructures with dose from 0.1 to 1.0 dpa involved defect cluster migration, with mutual elastic interactions often leading to spatial inhomogeneities and loop reactions. The nanocrystalline W alloys demonstrated loop coalescence with larger average loop area but lower density than ultrafine W alloys [22]. Nevertheless, the above irradiation-induced defects can result in severe hardening and embrittlement, which remarkably limited the utilization of W alloys in the fusion reactor [23]. It is necessary to significantly improve the mechanical properties and mitigate embrittlement of W-based materials.

Several techniques have been used to improve the ductility and toughness of W alloys, including micro alloying (Re [24] and Ta [25,26] etc.), dispersion of nanosized particles [27] and thermo-mechanical treatment [28]. Especially, the addition of a second precipitate was regarded as the most effective approach to form particle dispersion-strengthened tungsten maintaining higher strength and better ductility [29,30]. A series of results have demonstrated that dispersing fine oxide particles such as Y_2_O_3_ [31], La_2_O_3_ [32], TiC [33] and ZrC [34] in the W matrix can improve the toughness though strengthening the grain boundaries and increasing the recrystallization temperature. According to the previous research, we used the azeotropic distillation method combined with powder metallurgy method to fabricate W composites reinforced with c-ZrO_2_ [35]. Further, through adjusting ZrO_2_ content and swaging process to refine grain size and strengthen matrix, high strength and ductility W-ZrO_2_ materials can be achieved. However, few investigations about irradiation effect on the particle dispersion-strengthened W composites at present. The evolution mechanism of the ceramic phase during irradiation has not been systematically studied. In addition, the interaction mechanism between irradiation-induced defects and dispersed precipitate is still unclear. The mechanical properties of the particle dispersion-strengthened W composites after irradiation need to be further examined. 

The key objective of this study is to investigate microstructure evolution and mechanical properties by Au^+^ irradiation up to 100 dpa on W-ZrO_2_ composites and conventional pure W samples. The influence mechanism of ZrO_2_ phases on the number densities and size of irradiated defects are examined. Meanwhile, the sink strength of grain boundaries and precipitate interfaces for irradiated defects are quantitatively compared and analyzed. In addition, the impact of high dpa Au^+^ irradiation on the microstructure evolution and interfacial diffusion behavior of the ZrO_2_ phase are systematically revealed. It is highly expected to further clarify the relationship between irradiated defects and mechanical properties in W and W-ZrO_2_ composites. This work provides comprehensive and detailed analysis on irradiation behavior of W and W-ZrO_2_ composites, thereby giving insight into the potential of W-based plasma facing materials for nuclear applications.

## 2. Experimental

Three kinds of tungsten samples were used in this study. The first grade was pure W alloy with the grain size of 30~40 μm. The second grade was sintered W-1.5ZrO_2_ composites with grain size of ~20 μm and the diameter and volume fraction of ZrO_2_ precipitate were ~1.5 μm and ~5%, respectively. The third grade was swaged W-1.5ZrO_2_ composites with grain size of ~10 μm and the diameter and volume fraction of ZrO_2_ precipitate were ~0.5 μm and ~5%, respectively. The microstructure and mechanical properties of pure W, sintered and swaged W-1.5ZrO_2_ composites were compared in order to study the effect of grain size and ZrO_2_ precipitate with different sizes on the irradiation mechanism of tungsten composites. The corresponding microstructure of pure W and W-1.5ZrO_2_ composites were shown in Figure 1. Electron Back-Scatter Diffraction (EBSD) Inverse Pole Figure (IPF) maps indicated grain size of the three tested samples, and the black line represented grain boundaries. In addition, the red regions of phase maps displayed W phase, while the blue regions corresponded to the ZrO_2_ phase, showing the volume fraction and distribution of the ZrO_2_ phase. 

The three grade samples were prepared as the gauge of 2 mm × 2 mm × 1 mm. The samples were annealed at 1673 K for 2 h in argon atmosphere to eliminate the defects and internal stress introduced during the previous preparation process. The heavy ion irradiation was performed in the Sichuan University of Key Laboratory of Radiation Physics and Technology of Ministry of Education. The samples were irradiated with 3 MeV Au ions at room temperature and the irradiation fluence was up to 1.5 × 10^15^ Au^+^/cm^2^. The depth distribution of the damage and the implanted Au atoms were determined using the Stopping and Range of Ions in Matter (SRIM) code, assuming a displacement energy of 90 eV for tungsten element. The used simulation mode was the quick Kinchin-Pease mode. The predicted SRIM profiles were shown in Figure 2. From Figure 2, the maxima dpa at 200 nm depth was up to 100 dpa. The depth range of 50~200 nm from surface was used to characterize dislocations loops distribution in order to avoid surface effect and the influence of injected Au ion. Meanwhile, ZrO_2_ precipitate near surface was used for analyzing microstructure evolution after Au^+^ ion irradiation.

TEM specimens after Au ion irradiation were prepared by Focused ion beam (FIB) method on Thermo Fisher Scientific Scios 2. The samples were thinned up to about 100 nm in thickness for dislocations observation and quantification. Transmission electron microscopy (TEM) and energy dispersive X-ray spectroscopy (EDS) investigations were conducted on Tecnai G2 F30 TEM and Titan Themis Z spherical aberration corrected TEM. Determination of the loop Burgers vectors and the different type dislocations area and density percentages of three irradiated samples were performed in the same quantification zone using two beam bright field TEM conditions near the [001] zone axis (both <110> and <002> g vectors were used), referring to the works by Yi and El-Atwani et al. [18,21]. The invisibility criteria g·b = 0 of dislocation loops was used in this study, where g is the diffraction vector and b is the Burgers vector of the loop. In addition, for dislocation loop counting, circles were drawn in every image and the number of loops in every circle was counted using Image J. Meanwhile, to minimize quantification error in measuring the total density of dislocations, multiple regions were characterized. Quantification error can also occur due to image forces, glissile prismatic loop glide to the surface and foil thickness changes.

Surface hardness measurement after Au ion irradiation was performed at room temperature by Aligent 200 nanoindenter. All the tests were performed in continuous stiffness measurement mode with a constant loading rate $\dot{P}$/P = 0.05 s^−1^. For good statistical analysis, each sample was indented with at least 16 indents randomly, and the average value of the results was used in the analysis. Hardness was measured as a function of depth from the point of contact of the nanoindenter with the surface to a depth of about 2000 nm. Since it is well known that simply reporting and analyzing the measured nanohardness at a single depth is not acceptable, at a minimum, the Nix-Gao method can be used for obtaining an estimate of the bulk equivalent hardness in this study.

## 3. Results and Discussion

### 3.1. ZrO_2_ Phase Evolution

Au ion irradiation at room temperature has a significant impact on the ZrO_2_ precipitate. The microstructures after irradiation in the sintered and swaged W-1.5ZrO_2_ composites are almost the same. Therefore, the typical low magnification morphology of ZrO_2_ precipitate in the swaged W-1.5ZrO_2_ composites after 1.5 × 10^15^ Au^+^/cm^2^ radiation is shown in Figure 3a. It can be observed that there is an amorphous layer with a thickness of about 8 nm on the surface of the ZrO_2_ precipitate, labeled by a blue line in Figure 3b. Figure 3b demonstrates the atomic lattice near the surface of the ZrO_2_ precipitate has been damaged to a certain extent. A large number of striped or blocked irradiation damage regions (marked by a white dashed line) with about 10 nm in width are uniformly distributed and extended into the depth of 30~50 nm. As the irradiation depth increases, polycrystallization occurs in the second precipitate of ZrO_2_, whose thickness is approximately 200 nm. Figure 3c displays high magnification STEM-HAADF image of irradiation induced nanocrystalline grain boundaries with little different crystallographic orientation, indicating the grain boundary between nanocrystalline could be low angle grain boundary. The corresponding diffraction pattern of the polycrystallization region (marked by a green box) shows the diffraction scattering spot with crystal orientation, not the Debye ring, confirming this result as well. Figure 3d shows the selected area electron diffraction (SAED) pattern and high-magnification STEM-HAADF image of the red box region on the ZrO_2_ precipitate. With the irradiation depth further increasing, complete atomic arrangement of the ZrO_2_ precipitate can be seen. The SAED pattern can be indexed as only the ZrO_2_ phase, which has a face-center-cubic structure with a lattice parameter of 0.5128 nm. This implies that the impact of Au^+^ ion irradiation has basically disappeared. 

Figure 4 displays the EDS elemental mapping and line profile of irradiated (a and b) and unirradiated (c and d) phase boundary regions between the ZrO_2_ phase and the W matrix. The irradiated (100 nm depth) and unirradiated region (500 nm depth) are labeled as A and B in Figure 3a. As seen in the three elemental mapping, the ZrO_2_ phase and W matrix are located on the left and right of Figure 4a,c (the line profile direction is marked by red arrows). Simultaneously, it can be discovered that the phase boundary width of irradiated region is larger than the unirradiated region. These two region line profiles also demonstrate that the transition layer of different element components at the phase interface between the second phase ZrO_2_ and the W matrix increases from 2 nm to 8 nm. This suggests high dpa heavy ion irradiation aggravated the diffusion behavior of elements on the phase boundary. High-energy particle bombardment will give element solutes much greater migration energy in the ZrO_2_ precipitate and the W matrix. These high-energy alloying elements will escape from damage regions to low energy precipitate interfaces, even diffusion into other grain beyond the precipitate boundary. Hereafter, with the increase in irradiation damage, more element solutes in the W matrix and the ZrO_2_ precipitate diffuse each other and form an element transition layer with a certain width at the precipitate interfaces [2]. 

### 3.2. Dislocation Loops Characterization

Figure 5 shows the dislocation loops distribution of three irradiated samples in different depths (100 nm and 200 nm) after 1.5 × 10^15^ Au^+^/cm^2^ irradiation. According to statistical analysis, the size and density of dislocation loops around 100 nm depth within the pure W, sintered W-1.5ZrO_2_ composites and swaged W-1.5ZrO_2_ composites are 8.98 nm and 3.48 × 10^22^ m^−3^, 6.53 nm and 2.9 × 10^22^ m^−3^ and 5.45 nm and 2.63 × 10^22^ m^−3^, respectively. In the depth of 200 nm, the diameter and number of dislocations are 10.66 nm and 5.12 × 10^22^ m^−3^, 8.23 nm and 4.0 × 10^22^ m^−3^ and 7.94 nm and 3.86 × 10^22^ m^−3^, respectively. It can be concluded that as the irradiation depth increases from 100 nm to 200 nm, with the dpa value increasing from about 80 up to 100, the irradiation defects density, obviously, increases. In the irradiation damage peak regions (200 nm), dislocation loops entangle, coalesce and further grow up. By comparison, the dislocations in the W-1.5ZrO_2_ composites are smaller in size and less in number than pure W. Concurrently, the swaged W-1.5ZrO_2_ composites have the least and smallest dislocation loops among the three samples owing to the addition of ZrO_2_ precipitates and the increase in grain boundary density. In the W-based materials, the common Burgers vector of dislocation loops are 1/2<111> and <001>. Thereby, the statistical dislocations distribution using only one g vector condition can be inaccurate in this part. The following section will discuss the detailed information on dislocations loops in the three samples with both <110> and <002> g vectors. 

Figure 6 demonstrates the compared TEM bright images around 150 nm depth among three samples irradiated at room temperature with 100 dpa Au^+^ irradiation. These TEM images are using two-beam conditions near [100] axis zone, in which 6a, 6b and 6c are used by g vector <110>, while 6d, 6e and 6f are used by g vector <002>, respectively. Figure 7a–c show average diameter, density and average area of different type dislocation loops (including 1/2<111> and <001>) within the three samples after irradiation, respectively. Figure 7d summarizes total irradiation damage statistics among the above three grade W-based materials. As shown in Figure 6a, high density of dislocation loops with larger size can be seen in the irradiated pure W. By explicit counting and analysis, the number of dislocation loops with 1/2<111> Burgers vector is 3.72 × 10^22^ m^−3^, while the number of dislocation loops with <001> Burgers vector is 2.89 × 10^22^ m^−3^. Meanwhile, it can be concluded that the average diameter and calculated area of 1/2<111> loops are 9.53 nm and 68.63 nm^2^, which are much higher than <001> loops. However, with the addition of ZrO_2_, the dislocation density, obviously, decreases and loops size becomes smaller in the sintered W-1.5ZrO_2_ composites after irradiation. Concurrently, the density of 1/2<111> dislocation loops also evidently decrease to 3.16 × 10^22^ m^−3^. The average diameter and area of 1/2<111> loops and <001> loops exhibit substantially similar values of about 8.25 nm and 52.60 nm^2^. In the swaged W-1.5ZrO_2_ composites, the total dislocation loops density further decreases down to 5.68 × 10^22^ m^−3^, which is 1.3 times lower than pure W. Especially, both density and size of 1/2<111> loops (2.52 × 10^22^ m^−3^ and 7.22 nm) are significantly less than <001> loops (3.11 × 10^22^ m^−3^ and 7.26 nm), as shown in Figure 7b,c. 

Based on the dominant mechanism controlling defect evolution previously [36,37], it was reported that five distinct temperature ranges was defined for irradiation damage in W composites. In Stage II (100–700 K), recovery of tungsten occurs where interstitial migration occurs and growth of small interstitial loops to TEM resolvable size. In our study, the room temperature irradiation experiment corresponds to Stage II. Under high dpa ion irradiation, the density of dislocation loops increases and then tends to saturation. Afterwards, the dislocation loops can glide and coalesce with each other. In the pure W samples, considering surface effect and grain boundaries can sink defects to some extent, more 1/2<111> dislocation loops are glissile, and their coalescence is always expected at room temperature. Thereby, their sizes have to be larger than the <100> loops that are relatively sessile, as shown in Figure 7b. While the ZrO_2_ precipitates are added into the W matrix resulting in the decrease in grain size and increase in grain boundaries density. Simultaneously, ZrO_2_ precipitates with uniformed distribution can offer more specific surface areas to facilitate to sink the irradiated defects in the W-1.5ZrO_2_ composites. The small 1/2<111> dislocation loops are highly mobile, and, therefore, a large number of 1/2<111> dislocation loops are inclined to glide around precipitate and grain boundaries to annihilate rather than coalescence. By comparison, <100> loops are more stable to accumulate and grow within the grain interior. This leads to the total amount and average diameter of 1/2<111> dislocation loops decreasing to the level in the sintered W-1.5ZrO_2_ composites, which is almost the same as <100> loops. Combined with previous microstructure of swaged W-1.5ZrO_2_ sample, the proportion of the ZrO_2_-precipitate interface and grain boundaries is further increased, which can produce more preferential sink sites for irradiated dislocations. As expected, the number of glissile 1/2<111> loops, clearly, decrease to 2.52 × 10^22^ m^−3^ in the swaged W-1.5ZrO_2_ composites, which is over an order lower than pure W, as shown in Figure 7a. By comparison with the loop migration between 1/2<111> and <100> dislocation loops, the Burgers vector of 1/2<111> dislocation is parallel with (111) slip plane in the tungsten materials, indicating 1/2<111> dislocation can migrate in one-dimensional (1D) random walks by a thermally activated glide with a very low activation energy. Meanwhile, the Burgers vector of <100> loops is considered to be sessile or immobile since its glide direction is perpendicular to the Burgers vector [38]. Moreover, the activation energy for dislocation motion is significant correlation with temperature and weakly dependent on size of dislocation loops. Yi et al. have demonstrated that the corresponding average effective activation energy of 1/2<111> and <100> dislocation loops are 0.02 eV and 1.78 eV at room temperature, respectively [39]. According to the nature of dislocation loops, many more glissile 1/2<111> loops migrate to annihilate preferentially at precipitate interfaces and grain boundaries, resulting in the lower density of irradiated dislocations in the swaged W-1.5ZrO_2_ sample.

Normally, grain boundaries and the interface between the matrix and second precipitate are thought to be better traps for absorbing defects. The sink strength (number of defect sinks per unit area) of grain boundaries and ZrO_2_ precipitate interfaces for both regions can be calculated using the following equations [40,41]:(1)$$S_{b}=\frac{15}{h^{2}}$$
(2)$$S_{i}=4\pi N(0.3A){}^{1/2}$$
where S_b_ is the sink strength contributed from grain boundaries, h is the average grain size, S_i_ is the sink strength contributed from ZrO_2_ precipitate interfaces, N is the number density of ZrO_2_ precipitate, A is the surface area of the ZrO_2_ precipitates =, A = 4πr^2^ and r is the average diameter of the ZrO_2_ precipitate. Based on the microstructures statistics of three grade samples, the sink strength of grain boundaries S_b_ of pure W, sintered and swaged W-1.5ZrO_2_ samples are ~1.6 × 10^13^ m^−2^, ~3.7 × 10^13^ m^−2^ and ~1.5 × 10^14^ m^−2^, respectively. S_b_ for the swaged W-1.5ZrO_2_ sample is almost an order higher than S_b_ for pure W. This indicates that the addition of ZrO_2_ can refine grain and the swaged process can further reduce grain size to improve sink energy. Besides, we also discover the dislocations distribution around the grain boundaries and precipitate interfaces within the three samples, as shown in Figure 8. Although the dislocation-free zone could not be observed in Figure 8a–c, it can be confirmed that the dislocations density around grain boundaries is much lower than grain interior. By comparison, the loops density around 50 nm from grain boundaries is 2.86 × 10^22^ m^−3^, while the density at 200 nm is 3.34 × 10^22^ m^−3^ in irradiated pure W. Similarly, within the swaged W-1.5ZrO_2_ sample, the dislocation density at 50 nm from grain boundaries is 1.53 × 10^22^ m^−3^, which is much lower than that at 200 nm (2.78 × 10^22^ m^−3^). Except for grain boundaries, the precipitate interfaces between the W matrix and the ZrO_2_ precipitate can offer strong sink energy alike. According to Equation (2), S_i_ is ~7.8 × 10^14^ m^−2^ for the swaged W-1.5ZrO_2_ sample, over two times higher than S_i_ for the sintered W-1.5ZrO_2_ sample, which is ~3.6 × 10^14^ m^−2^. As shown in Figure 8d,e, the dislocations-denuded zones appear along the precipitate interfaces, whose widths are ~25 nm and ~40 nm in the sintered and swaged W-1.5ZrO_2_ samples (marked by green and red dashed lines, respectively). Meanwhile, the dislocation density at 50 nm around precipitate interface are 9.42 × 10^21^ m^−3^ and 7.88 × 10^21^ m^−3^ and, correspondingly, increase up to 2.65 × 10^22^ m^−3^ and 2.58 × 10^22^ m^−3^ at 200 nm in the sintered and swaged 1.5ZrO_2_, respectively. The formation of denuded zones and the remarkable decline of dislocations density both demonstrate that precipitate interfaces can effectively absorb and annihilate irradiated dislocation loops. Furthermore, the swaging process can increase precipitate interfaces fraction to enhance their sink strength. Combined with the above analysis, it can be noteworthy that the effect of sinking irradiated dislocations at the precipitate interface is, obviously, far superior to that at the grain boundary. Overall, refining grain size by the thermal-mechanical process and tailoring the size and distribution of the ZrO_2_ precipitate can both strongly improve the sink strength of interfaces and the boundaries absorbing irradiated defects, thus enhancing irradiation resistance properties of W-based materials. 

### 3.3. Mechanical Behavior

Nanoindentation measurement can be a useful monitor to determine the change in surface mechanical properties of the ion-irradiated samples. Figure 9a,b show representative load-displacement and hardness displacement data of three W-based materials after 1.5 × 10^15^ Au^+^/cm^2^ irradiation. On the whole, three samples present that with the increase in measured depth, the applied load gradually increases. Since it is well known that the indentation hardness is sensitive to near surface regions, we ignore the data in depths smaller than 50 nm. As for the data in depths larger than 50 nm, the nanohardness decreases slightly with penetration depth and tends to be stable, almost the same as the matrix hardness values, as shown in Figure 9b. This can be explained by the Nix-Gao model [42], which predicts the hardness-depth profile by the following equation:(3)$$H=H_{0}{\left(1+\frac{h^{*}}{h}\right)}^{\frac{1}{2}}$$
where H_0_ is the hardness at infinite depth, h* is a characteristic length which depends on the material and the shape of the indenter tip. For a given material and indenter, the parameter h* can be considered to be a constant. According to the Nix-Gao method reported recently by Kasada et al. [43,44], the irradiation hardening of three samples can be better evaluated by plotting the nanohardness data as H^2^ versus h^−^^1^, as shown in Figure 9c. As for the irradiated samples, the data show the presence of bilinearity with a shoulder (h_c_) at depth around 150 nm among three W-based materials. The estimated hardness at h_c_ depth (~150 nm) can be considered to the relatively accurate evaluation of irradiation hardness. Figure 9d shows the hardness increment of three irradiated W-based samples compared to the unirradiated samples. The hardness increase values of pure W, sintered and swaged W-1.5ZrO_2_ composites show 8.67 Gpa, 5.54 GPa and 4.52 GPa, respectively. It can be concluded that the pure W has the highest irradiation hardening effect; in contrast, the swaged W-1.5ZrO_2_ composites have the lowest effect. This indicates that the swaged W-1.5ZrO_2_ composites achieve the best irradiation hardening resistance properties.

In order to quantify the contribution of irradiated induced defects on hardening of the three samples in this study, the measured nanoindentation hardness is compared in the following with the predictions from the well-known dispersed barrier-hardening (DBH) model [45]. The DBH model is given in Equation (4):(4)$$\Delta \tau =\alpha \mu b\sqrt{\mathrm{Nd}}$$
where Δτ is the change in the resolved shear stress, α is the barrier strength (α~0.4 for dislocation loops), and µ is Tungsten Shear Modulus = 161 GPa. Previously, b is the module of the Burgers vector in the tungsten. N and d are total volumetric defects density and average diameter of irradiated samples, respectively. In this study, in order to increase calculation accuracy, we decide to choose previous analytical statistics (diameter and density) on 1/2<111> and <001> dislocation loops and then compute the corresponding shear stress change Δτ_<111>_ and Δτ_<001>_, respectively. The total shear stress change Δτ_total_ is shown in Equation (5). Hereafter, the uniaxial yield strength Δσ can then be obtained by Equation (6).
(5)$$\Delta \tau {}_{\mathrm{total}}=\sqrt{\Delta \tau_{<111>}^{2}+\Delta \tau_{<001>}^{2}}$$
(6)$$\Delta \sigma =M\Delta \tau_{\mathrm{total}}$$
where M is the Taylor Factor = 3.06 (for equiaxed BCC metals). Finally, the hardness increment using by DBH model is assumed to be three times the uniaxial yield strength [46]. Table 1 shows the comparison between the calculated hardness changes $\Delta H_{c}$ and the experimentally obtained hardness changes $\Delta H_{e}$ for the three irradiated W-based samples. It can be seen that the calculated hardness changes using the DBH model are almost consistent with hardness changes obtained by a nano-indentation measurement. Overall, it appears that the dislocation loops configuration and geometric properties are the main contributors for the irradiation-induced hardening effect in the ion-irradiated W-based materials at room temperature. Based on previous microstructural analysis, it is not surprising to reach this conclusion. The pure W alloys have coarse grain size and the sink strength of grain boundaries are limited, leading to the accumulation of irradiated defects, therefore, the irradiation hardening effect is severely enhanced. Nevertheless, in the W-ZrO_2_ composites, the high density of grain boundaries and precipitate interfaces between the ZrO_2_ and the matrix can effectively absorb a large number of small mobile dislocation loops to delay or inhibit the irradiation hardening effect, especially in the swaged W samples. Hence, the swaged W samples have the lowest change in hardness after high dpa Au^+^ irradiation, shown in Table 1. Therefore, the swaged W-1.5ZrO_2_ samples can be the most promising candidate PFMs for nuclear fusion reactor, maintaining a better anti-radiation hardening performance.

## 4. Conclusions

In this study, the microstructure evolution and mechanical properties of pure W, sintered and swaged W-1.5ZrO_2_ composites have been systematically investigated after 1.5 × 10^15^ Au^+^/cm^2^ irradiation at room temperature. The main results of this research are summarized as follows:(1)The spatial structure of ZrO_2_ phase after high dpa Au^+^ irradiation consists of three layers, including the amorphous layer (~8 nm), polycrystallization region (~200 nm) and complete ZrO_2_ matrix. Meanwhile, the element transition layer width increase from ~2 nm up to ~8 nm, attributing to diffusion aggravation at phase boundaries by radiation.(2)The average diameter and density of the different Burgers vectors (1/2<111> and <001>) dislocation loops are quantitatively determined in three irradiated samples. With the addition of the ZrO_2_ precipitate, the total number and size of dislocations significantly decrease, especially 1/2<111> loops. Many more glissile 1/2<111> loops migrate to annihilate around the precipitate interface, with a higher sink strength of 7.8 × 10^14^ m^−2^.(3)Owing to the high density of the precipitate interfaces and grain boundaries, the irradiation hardening effect of the swaged W-1.5ZrO_2_ composites severely weakened after high dpa Au^+^ irradiation, resulting in the lowest change in irradiation hardness of 4.52 GPa, which is almost consistent with calculation values.

## Figures and Tables

**Figure 1 materials-15-01985-f001:**
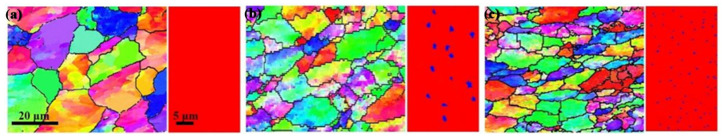
EBSD IPF and phase maps of three tested samples: (**a**) pure W, (**b**) sintered W-1.5ZrO_2_ composites, (**c**) swaged W-1.5ZrO_2_ composites.

**Figure 2 materials-15-01985-f002:**
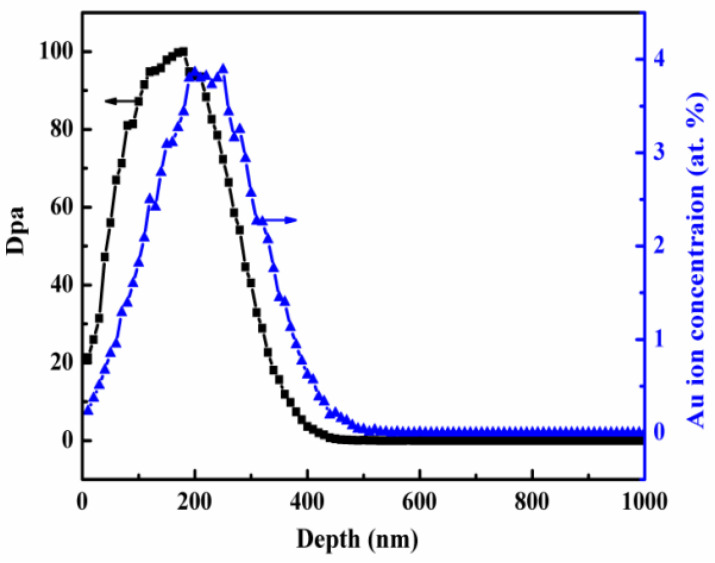
The predicted SRIM profile of the displacement damage and implanted Au ion concentration.

**Figure 3 materials-15-01985-f003:**
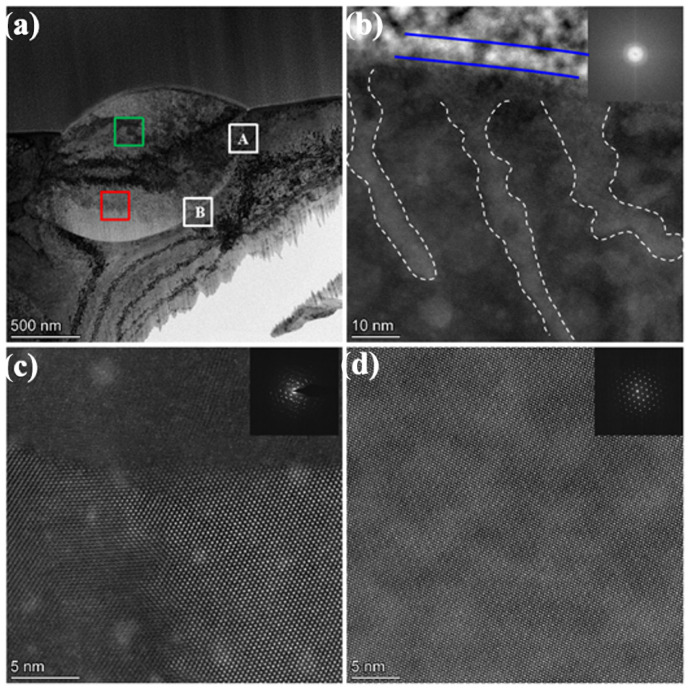
The typical morphology of ZrO_2_ precipitate in the swaged W-1.5ZrO_2_ composites after 1.5 × 10^15^ Au^+^/cm^2^ radiation: (**a**) low magnification TEM images, (**b**) STEM-HAADF atomic lattice image on the ZrO_2_ precipitate surface and corresponding FFT diffraction patterns, (**c**) STEM-HAADF image of polycrystallization regions and corresponding SAED patterns, (**d**) STEM-HAADF image of unirradiated regions and corresponding SAED patterns. A and B in the white box represents irradiated and unirradiated region, respectively.

**Figure 4 materials-15-01985-f004:**
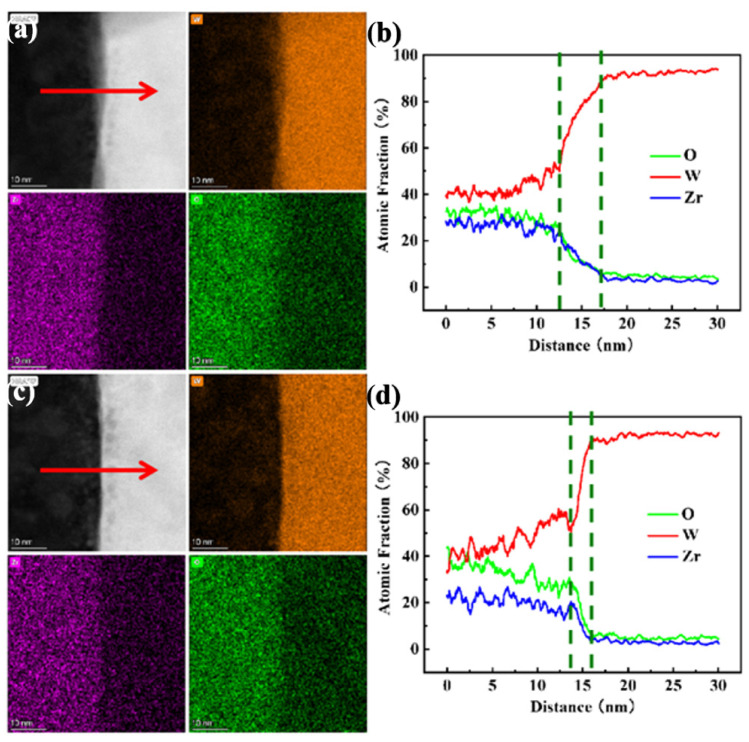
The EDS elemental mapping (W, Zr, O) and line profile of irradiated (**a**,**b**) and unirradiated (**c**,**d**) phase boundary regions between the ZrO_2_ phase and the W matrix.

**Figure 5 materials-15-01985-f005:**
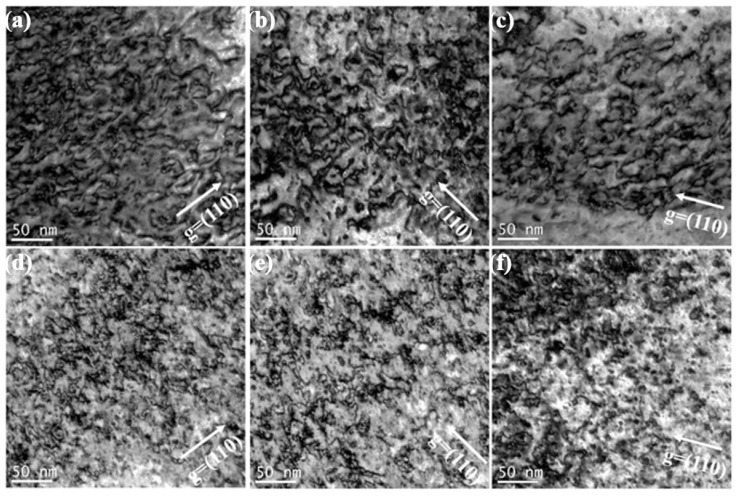
The dislocation loops distribution of three grade samples in different depth with g vector of <110> after 1.5 × 10^15^ Au^+^/cm^2^ irradiation: (**a**–**c**) corresponds to 100 nm, (**d**–**f**) corresponds to 200 nm.

**Figure 6 materials-15-01985-f006:**
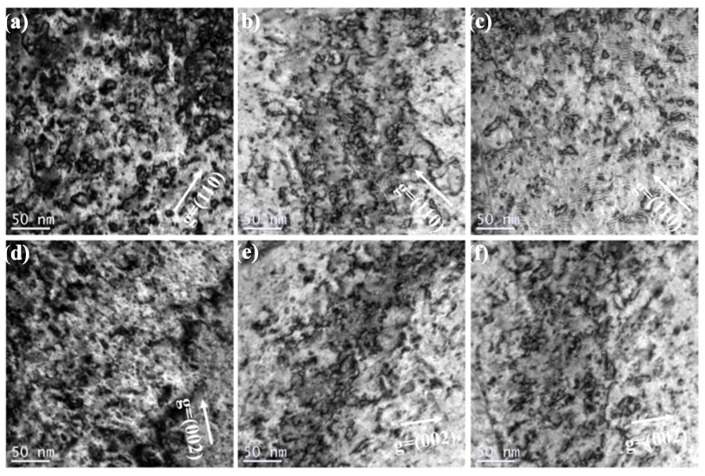
Bright-field TEM images using two beam conditions from the [001] zone axis showing loops in three irradiated W-based materials: (**a**) pure W, (**b**) sintered W-1.5ZrO_2_ composites, (**c**) swaged W-1.5ZrO_2_ composites with g vector of <110>, (**d**–**f**) corresponding to above three samples with g vector of <002>, respectively.

**Figure 7 materials-15-01985-f007:**
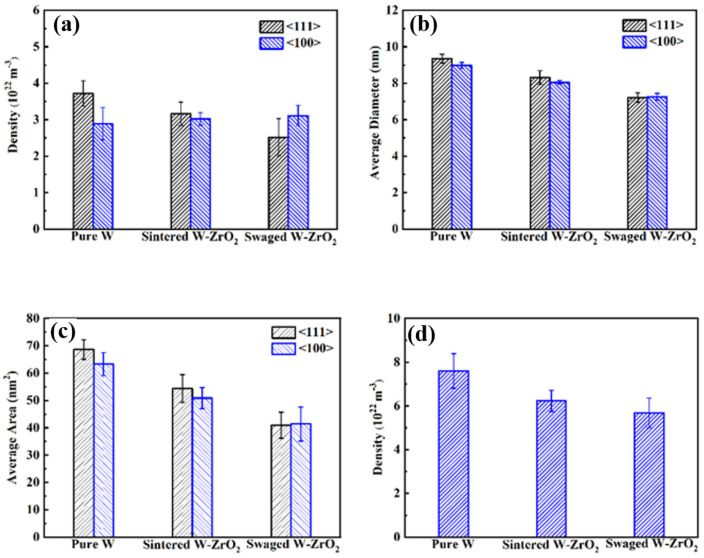
Damage statistics for the three irradiation samples after 1.5 × 10^15^ Au^+^/cm^2^ irradiation: (**a**) dislocation density of <111> and <001> type loops, (**b**) average diameter of <111> and <001> type loops, (**c**) average area of <111> and <001> type loops, (**d**) total dislocation loops density.

**Figure 8 materials-15-01985-f008:**
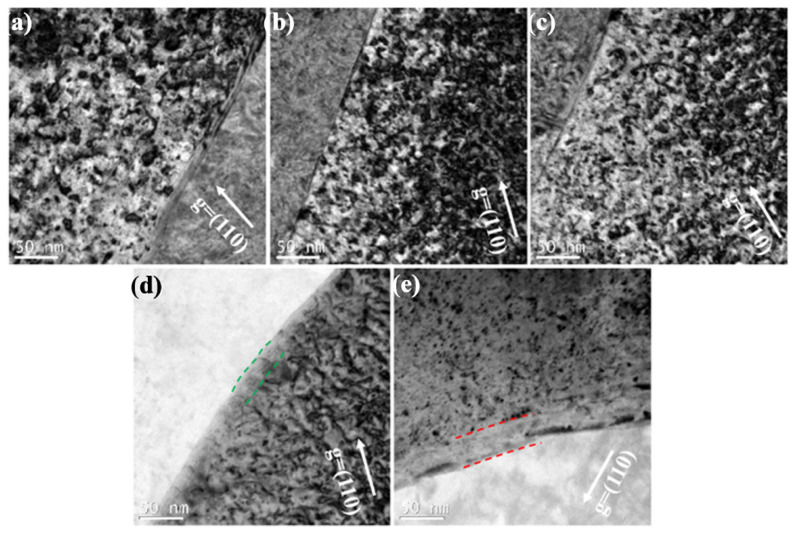
The dislocations distribution around the grain boundaries and precipitate interfaces within the three irradiated W-based samples: (**a**) pure W, (**b**) and (**d**) sintered W-1.5ZrO_2_ composites, (**c**) and (**e**) swaged W-1.5ZrO_2_ composites.

**Figure 9 materials-15-01985-f009:**
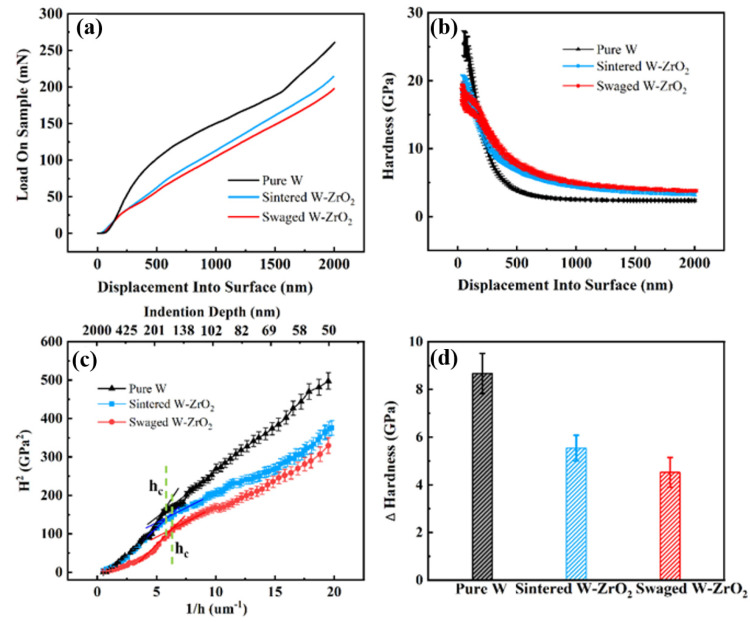
Irradiation hardening effect statistics of three irradiated samples: (**a**) load-displacement curves, (**b**) corresponding hardness-displacement curves, (**c**) the nanohardness curves as H^2^ versus h^−1^, (**d**) experimental hardness values at h_c_ depth (~150 nm).

**Table 1 materials-15-01985-t001:** The comparison between the calculated hardness changes $\Delta H_{c}$ and the experimentally obtained hardness changes $\Delta H_{e}$ for the three irradiated W-based samples.

Samples	$\mathrm{Calculated}\;\mathrm{Change}\;\mathrm{in}\;\mathrm{Hardness}\;\Delta H_{c}\;(\mathrm{GPa})$	$\mathrm{Experimental}\;\mathrm{Change}\;\mathrm{in}\;\mathrm{Hardness}\;\Delta H_{e\;}\;(\mathrm{GPa})$
Pure W	9.49	8.67
Sintered W-1.5ZrO_2_	6.32	5.54
Swaged W-1.5ZrO_2_	5.09	4.52

## Data Availability

Data is contained within the article.

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
