# Peer review of "Superior Radiation Resistance of ZrO2-Modified W Composites"

_materials, 2022, doi:10.3390/ma15061985_

Round 1

Reviewer 1 Report

This paper reports a study on the microstructure and mechanical properties of W and W-ZrO2 composites and on the impact of the addition of a ZrO2 phase on the irradiation resistance mechanism of W materials.

My first and most important comment is that the English has to be strongly improved to allow a correct evaluation of the scientific quality of the paper. Very often, the phrases construction is wrong and, for this reason, the meaning of authors’ statements is obscure.

Sincerely speaking in my opinion it is impossible to evaluate the paper in the present form.

Despite this, I would like to report some errors:

  • While reporting values, the sigmas are never indicated.

  • Panels a and c of figure 7 report exactly the same picture, while they should be different, according to the figure caption.

  • All the fields starting from line 401 have not been filled by the authors.

Having in mind that we all make mistakes (incluing myself!), may I say that even controlling these details during a paper submission has not to be considered a secondary task? Reviewers make available their time to evaluate papers, to suggest changes and so on, not to act as proofreaders.  

Author Response

Response to Reviewer 1 Comments

This paper reports a study on the microstructure and mechanical properties of W and W-ZrO2 composites and on the impact of the addition of a ZrO2 phase on the irradiation resistance mechanism of W materials.

Point 1: My first and most important comment is that the English has to be strongly improved to allow a correct evaluation of the scientific quality of the paper. Very often, the phrases construction is wrong and, for this reason, the meaning of authors’ statements is obscure.

Response: Thank you for your suggestion. After carefully read the comments, we realize that the major merits of our work have been identified or recognized by the reviewers. The revised manuscript has already been improved.

Point 2:  Despite this, I would like to report some errors:

  • While reporting values, the sigmas are never indicated.

  • Panels a and c of figure 7 report exactly the same picture, while they should be different, according to the figure caption.

  • All the fields starting from line 401 have not been filled by the authors.

Response: Thanks for your valuable comments. Firstly, the sigma  indicates the uniaxial yield strength and the symbol  has been inserted into line 365 in the revised manuscript. Secondly, we have corrected Figure 7c in the revised manuscript. Figure 7a and c are the dislocation density and average loops area in the three tested W based materials, respectively. At last, the Author individual contributions has been filled in the revised manuscript.

Reviewer 2 Report

The paper presents an extensive irradiation study of W-ZrO2 composites for nuclear application. The results are suitable for the research community. 

- Some minor English check required. 

- Introduction, last paragraph: “… lattice defects such as dislocation loops, Frenkel pairs, and voids”, do the authours mean vacancies? Void is used for macroscopic defects.

Author Response

Response to Reviewer 2 Comments

The paper presents an extensive irradiation study of W-ZrO2 composites for nuclear application. The results are suitable for the research community. 

- Some minor English check required. 

- Introduction, last paragraph: “… lattice defects such as dislocation loops, Frenkel pairs, and voids”, do the authours mean vacancies? Void is used for macroscopic defects.

Response 1: Thank you for your good advice. According to your advice, we have revised English expression in the revised manuscript. Meanwhile, The “voids” has changed to “vacancies” in the last paragraph: “… lattice defects such as dislocation loops, Frenkel pairs, and voids” within the revised introduction. Meanwhile, English expression has already been improved.

Reviewer 3 Report

This paper is undoubtedly useful and interesting, but in this form it cannot be recommended for publication yet and some points need to be clarified more clearly.

  1. I would like to note that the introduction is written at a high scientific level.
  2. Nevertheless, please provide more information about the structure ZrO2.
  3. What happens to the ZrO2 oxide as a result of irradiation?
  4. Whether color centers are formed in it, as happens in other oxides, such as Al2O3: Popov, A.I.; Lushchik, A.; Shablonin, E.; Vasil’chenko, E.; Kotomin, E.A.; Moskina, A.M.; Kuzovkov, V.N. Comparison of the F-type center thermal annealing in heavy-ion and neutron irradiated Al2O3single crystals.  Instrum. Methods Phys. Res. Sect. B Beam Interact. Mater. At. 2018433, 93–97.
  5. When talking about loop migration, please provide more details about actual mechanism and corresponding defect mobility activation energy.

Author Response

Response to Reviewer 3 Comments

This paper is undoubtedly useful and interesting, but in this form it cannot be recommended for publication yet and some points need to be clarified more clearly.

Point 1: I would like to note that the introduction is written at a high scientific level.

Response: Thanks for your conductive advice. We have revised the introduction in the revised manuscript. The corresponding scientific detail has been inserted into the revised introduction.

Point 2: Nevertheless, please provide more information about the structure ZrO2.

Response: Thank you for your advice. The detailed information about the structure ZrO2 has added into the line 174-175 in the revised manuscript.

Point 3: What happens to the ZrO2 oxide as a result of irradiation

Response: Thanks for the good comment. The results of irradiation about ZrO2 oxide have shown in the part 3.1 in the original manuscript. In summary, the spatial structure of ZrO2 phase after high dpa Au+ irradiation consists of three layers, including amorphous layer (~8nm), polycrystallization region (~200nm) and complete ZrO2 matrix. Meanwhile, the element transition layer width increase from ~2nm up to ~8nm, attributing to diffusion aggravation at phase boundaries by radiation.

Point 4: Whether color centers are formed in it, as happens in other oxides, such as Al2O3: Popov, A.I.; Lushchik, A.; Shablonin, E.; Vasil’chenko, E.; Kotomin, E.A.; Moskina, A.M.; Kuzovkov, V.N. Comparison of the F-type center thermal annealing in heavy-ion and neutron irradiated Al2O3single crystals.  Instrum. Methods Phys. Res. Sect. B Beam Interact. Mater. At. 2018433, 93–97.

Response: Thanks for your comment. The corresponding experiment and analysis on the color centers have not been carried out. It is a novel idea and we will try to discuss this aspects in the futher research.

Point 5: When talking about loop migration, please provide more details about actual mechanism and corresponding defect mobility activation energy.

Response: Thank you for your suggestion. The discussion section on loop imgration mechanism and defect mobility activation energy has been added into line 280-291 in the revised manuscript.

Round 2

Reviewer 3 Report

The article has been well improved in accordance with the opinion of the reviewer and therefore can be recommended for publication as it is.